

# Level of attention mediates the association between connectedness to nature and aesthetic evaluations of photographs of nature

Neil Harrison

Department of Psychology, Liverpool Hope University, Liverpool, Merseyside, United Kingdom

## ABSTRACT

Aesthetic experiences of nature are associated with beneficial psychological and behavioural outcomes. We investigated in a laboratory study whether an individual's level of connectedness to nature is associated with their aesthetic sensitivity to images of natural scenes, and whether the amount of attention allocated to the images mediated this association. Participants ($N = 82$) viewed 14 photographs depicting natural scenes and evaluated them on three aesthetic dimensions and completed the Connectedness to Nature (CN) and Openness to Experience (OtE) scales. CN positively predicted pleasure, beauty and aesthetic emotion, independently of OtE. The amount of attention participants paid to the images mediated the relationship between connectedness to nature and aesthetic pleasure, and connectedness to nature and beauty ratings. These findings extend our understanding by showing that attention is an important mechanism through which nature connectedness influences aesthetic responses of pleasantness and beauty in response to natural scenes. The findings have real-world implications as appreciation of the aesthetic qualities of nature is associated with a number of beneficial psychological outcomes.

## INTRODUCTION

Natural scenes can evoke a wide range of aesthetic responses, from simple liking and preference to more complex emotional states such as awe and wonder (*Joye & Bolderdijk, 2015*; *Silvia et al., 2015*). These sensory pleasures and emotional responses are no doubt of intrinsic worth, but importantly they may also be associated with a number of positive psychological and behavioural outcomes. For example, several studies have shown that an appreciation of the aesthetic qualities of a natural environment influences outcomes such as tourist visits (*Awaritefe, 2004*), increases well-being and prosocial behaviours (*Capaldi et al., 2017*; *DeLucio & Mugica, 1994*; *Zhang, Howell & Iyer, 2014*; *Zhang et al., 2014*), and increases pro-environmental behaviours such as conservation and environmental citizenship (*Diessner et al., 2018*). Given the robust evidence demonstrating its broad importance, it is vital to extend our understanding of the psychological factors associated with aesthetic responses to nature.

Corresponding author
Neil Harrison, harrisn@hope.ac.uk

Empirical studies in environmental aesthetics have tended to investigate aesthetic preferences for different types of landscape, for example forests and savannahs (*Heerwagen & Orians, 1993*) and how such preferences are associated with the landscape's formal and structural properties (for reviews, see *Ulrich, 1983*; *Yang et al., 2021*). However, it is well-established that aesthetic preferences and experiences result from an interaction between stimulus properties and the way these properties are processed and integrated by the viewer (*e.g.*, *Jacobsen, 2006*). Factors as diverse as whether a landscape is viewed during a pandemic-related lockdown, to demographic characteristics of the viewer such as cultural background and socio-economic status, are known to influence landscape preferences (*e.g.*, *Felisberti & Harrison, 2022*; *Swanwick, 2009*; *Yu, 1995*). So far, though, there have been relatively few studies examining the role of psychological traits, such as personality and related person-centered constructs, in relation to aesthetic appreciation of the natural world (*Yang et al., 2021*).

Accounts of aesthetic experience that emphasise the importance of the psychological connection between the viewer and the object viewed provide the theoretical background to this study. In the environmental aesthetics literature, this psychological connection is fundamental in the engaged aesthetics account of *Berleant (1992)*, who proposed that the unity of perceiver and object is essential for the emergence of aesthetic experiences of nature. A similar emphasis can be found in theories of aesthetic responses to human-made artefacts. For example, positive aesthetic outcomes (such as being moved and flow-like states) are thought to occur when an artwork is congruent with some important aspect of the viewer's identity (*Pelowski et al., 2017*). Together, these theoretical views show that a sense of psychological relatedness, or connection, between the viewer and the object viewed may be an important factor governing aesthetic evaluations and experiences of nature.

Based on the above considerations, the starting point for the current study is that a person's psychological connectedness and affiliation with the natural world, reflecting the degree to which nature is incorporated into their sense of self, may have an important influence on their aesthetic appreciation of nature. A sense of connection with nature, commonly referred to as 'connectedness to nature', is a trait-like construct characterized by an emotional connection and attachment to the natural world, and by beliefs about a person's connection to the natural world (*Mayer & Frantz, 2004*; *Perrin & Benassi, 2009*). A key component of connectedness to nature is the degree to which nature is included within an individuals' cognitive representation of self (*Schultz, 2002*). Here we investigated whether connectedness to nature may influence an individual's aesthetic sensitivity to natural scenes.

There is an emerging literature showing that individual differences in connectedness to nature may influence a number of different perceptual and emotional responses evoked by viewing natural environments (*Berto et al., 2018*; *Clayton, 2021*; *Davis & Gatersleben, 2013*; *McMahan et al., 2018*; *Tang, Sullivan & Chang, 2015*). For example, *Berto et al. (2018)* showed that participants with higher levels of connectedness to nature reported higher levels of perceived restorative effects, and higher levels of preference, for natural landscapes. *McMahan et al. (2018)* reported that participants with higher levels of

connectedness to nature had more positive emotional outcomes (measured using both explicit and implicit measures of emotional state) from exposure to immersive simulations of natural and built environments. For visitors to both a wild natural landscape and a manicured environment, *Davis & Gatersleben (2013)* found that higher levels of connectedness to nature predicted more positive emotional experiences—both high and low arousal emotions (exciting and calming). They also found that those high in connectedness to nature had less negative or fearful experiences in more extreme/scary natural environments, *i.e.*, it may have provided a buffering effect against negative emotional responses. Evidence also supports the notion that people high in connectedness to nature may experience enhanced perceptual qualities (such as legibility, mystery, and restorative potential) of images of natural landscapes (*Tang, Sullivan & Chang, 2015*), and that individuals who are more connected to nature (measured using the Environment Identity Scale) may be more emotionally affected by images of degraded (*i.e.*, polluted) landscapes (*Clayton, 2021*). Despite the large variety of types of landscapes shown, taken together these studies provide evidence that connectedness to nature is a potentially robust predictor of a variety of perceptual and emotional responses evoked by viewing natural environments.

Existing evidence also shows that individuals with a stronger degree of connectedness to nature have a higher self-reported tendency to perceive natural beauty (*Harrison & Clark, 2020*; *Zhang, Howell & Iyer, 2014*). In these studies, higher levels of connectedness to nature were shown to be associated with increased engagement with natural beauty, as measured by the Engagement with Natural Beauty (ENB) scale (*Diessner et al., 2008*), which reflects a dispositional sensitivity to affective and physiological responses evoked by natural beauty. A limitation of these studies, though, was that their measurement of aesthetic experiences of nature involved using a memory-based reporting format where participants self-reported how they typically felt. Retrospective self-report is associated with a number of distinct biases (*Conner & Barrett, 2012*), therefore the current study aimed to avoid these limitations by measuring aesthetic responses in the moment.

Viewing natural scenes can elicit a number of different aesthetic experiences that vary in complexity and depth (*Silvia et al., 2015*). Our choice of measures was guided by the affect-space framework for aesthetic experience (*Schubert, North & Hargreaves, 2016*). The affect-space framework characterizes aesthetic experiences in terms of hedonic tone and aesthetic judgement. Here we measured responses reflecting shallow hedonic tone (pleasantness), deep hedonic tone (awe, being moved, profundity) and aesthetic judgement (beauty) to capture a range of aesthetic responses to the scenes.

Openness to experience (a personality trait defined by curiosity and receptivity to new experiences; *Costa & McCrae, 1992*) is a consistent predictor of aesthetic appreciation. Previous findings have shown that openness to experience is a robust predictor of aesthetic responsiveness to artworks (*Chamorro-Premuzic et al., 2009*; *McCrae, 2007*; *Palumbo et al., 2023*) and to music (*Colver & El-Alayli, 2015*), and studies have found that participants who scored higher in openness to experience reported greater engagement with natural beauty (*Harrison & Clark, 2020*; *Zabihian & Diessner, 2016*) and rated nature scenes as more emotionally evocative (*Silvia et al., 2015*). Therefore, in the current study it was

important to investigate the unique contribution of connectedness to nature on aesthetic experience, by controlling for participants' degree of openness to experience.

There is evidence to suggest that viewers with higher levels of connectedness to nature may allocate more attention to natural scenes, in other words that they show enhanced visual engagement with natural scenes during a viewing episode, compared to viewers with lower levels of connectedness to nature (*Bingjing, Chen & Shuhua, 2022*; *Wu et al., 2013*). *Wu et al. (2013)* recorded participants' eye movements (Experiment 2) while they viewed images of real-world scenes and found that participants higher in connectedness to nature visually explored the landscape scenes more thoroughly than participants lower in connectedness to nature. Similarly, *Bingjing, Chen & Shuhua (2022)* reported that, in a virtual environment, participants who scored higher in connectedness to nature spent more time looking at natural features, such as trees, within scenes, as indexed by both total visit duration and total fixation duration. Given that eye movements are informative about patterns of visual attention (*Rayner, 2009*), together these studies provide evidence that people higher in connectedness to nature may allocate increased levels of attention to natural scenes compared to people lower in connectedness to nature.

Moreover, empirical findings demonstrate an association between the amount of visual attention allocated to a natural scene (or indeed an artwork) and the evoked aesthetic response. For example, *van den Berg, Joye & Koole (2016)* showed that viewing time was positively correlated with a scene's perceived restorativeness (*i.e.*, its potential to evoke positive emotion and relaxation). Along similar lines, it has been shown that increased visual attention to artworks, as indexed by longer overall viewing times, was related to enhanced aesthetic evaluation of the artworks (*Brieber et al., 2014*; *Palumbo et al., 2023*). It seems likely that enhanced attention to the image allowed extraction of more detailed information about important features or compositional properties of the image, contributing to an enhanced evaluation of the image's aesthetic qualities. Here we measured attention to each image using a self-report scale which has been used in several previous studies (*Wu et al., 2013*; *Silvia et al., 2015*) and which has been shown to be closely correlated with more objective measures of attention derived from eye tracking (*Wu et al., 2013*).

Recently, *Palumbo et al. (2023)* showed that visual attention (measured by dwell time) mediated the relationship between a personality trait (openness to experience) and aesthetic responses to artworks. Greater visual attention to the artworks in those higher in openness to experience may have allowed them to extract more information about the compositional properties of the artwork, resulting in an enhanced aesthetic response. Based on the conceptual similarity with the above mediation model, we reasoned that in the case of viewing images of nature, a viewer's level of attention to the image might explain the relationship between the viewer's level of connectedness to nature and their aesthetic experience of the image. We investigated this by testing whether participants' self-reported level of attention mediated the relationship between connectedness to nature and the aesthetic experiences evoked by the images.

The current study aimed to investigate the relationship between connectedness to nature and participants' aesthetic experiences evoked by viewing photographs of natural

scenes, and in particular, to test the role of attention as a mediator in this relationship. Firstly, we predicted that participants higher in connectedness to nature would rate images of natural scenes as more pleasing and beautiful and give higher ratings of aesthetic emotion, compared to participants lower in connectedness to nature, while controlling for openness to experience. Secondly, we predicted that connectedness to nature would be positively related to the level of attention that participants paid to the images. Thirdly, we predicted that the level of attention paid to the scenes would mediate the association between connectedness to nature and aesthetic responses. We extend previous research in several ways. Firstly, we measured three aspects of participants' aesthetic experiences, namely pleasure, beauty and aesthetic emotions. Further, we investigated whether connectedness to nature uniquely predicted aesthetic experiences, by controlling for openness to experience (a trait that strongly predicts aesthetic responsiveness). Finally, we investigated whether the association between connectedness to nature and aesthetic evaluations would be mediated by the amount of attention that participants paid to the images.

## MATERIALS AND METHODS

### Participants

Eighty-two participants took part in the study. Participants were aged between 18 to 42 years ($M$ = 21.4; $SD$ = 3.9 years), and there were 63 women and 19 men. The study was conducted in accordance with the standard ethical guidelines as defined in the Declaration of Helsinki, and written informed consent was obtained prior the study. The study was approved by the Ethics Committee of the Faculty of Science at Liverpool Hope University (Ethics approval number: S-16-11-2017 NJ 003).

### Stimuli and measures

#### Images of nature

Fifteen colour photographs depicting natural scenes (*e.g.*, waterfalls, mountains, clouds, rivers, sea) were selected from the Pexels online photo database (http://www.pexels.com). Each image was displayed onscreen with a size of 24 cm × 15 cm. All of the pictures and the study data can be viewed at this study's Open Science Framework project (https://doi.org/10.17605/OSF.IO/WPXCN).

#### Connectedness to nature

As previously described in *Harrison & Clark (2020)*, emotional connection with the natural world was measured with the 14-item Connectedness to Nature Scale (*Mayer & Frantz, 2004*; the authors have permission to use this instrument from the copyright holders). This 14-item scale (*e.g.*, "I think of the natural world as a community to which I belong") is measured on a five-point Likert scale ranging from 1 (strongly disagree) to 5 (strongly agree). A summed score was derived for each participant with higher scores reflecting stronger connectedness to nature. *Mayer & Frantz (2004)* provided evidence of its reliability (α = 0.84) and validity (*e.g.*, positive correlations with environmental concern). The scale displayed a good level of internal consistency in the current sample (α = 0.83).

### Openness to experience

The Openness/Intellect subscale from the Big Five Aspect Scales (BFAS) was administered (*DeYoung, Quilty & Peterson, 2007*; the authors have permission to use this instrument from the copyright holders). This subscale measures two distinct, but related, traits: openness to experience and intellect. In the analyses reported here, only the openness trait was included, as this trait reflects aesthetic receptivity and feelings (*DeYoung, Grazioplene & Peterson, 2012*). A good level of internal consistency was observed in the current sample ($\alpha = 0.71$).

### Self-report items

A series of seven self-report items measured participants' responses to each image. Participants were asked "Did you find this picture…" followed by the seven items. Pleasing was measured with one item, and beauty was measured with one item. Aesthetic emotions were measured with a 4-item aesthetic emotions scale which included items for awe-inspiring, moving, profound and boring (reverse-scored). In the current sample the scale displayed a good level of internal consistency ($\alpha = 0.79$). Participants responded to each item using a 7-point rating scale (1 = not at all, 7 = very much). A further item asked: "How closely were you paying attention to the picture?" (1 = not at all, 7 = very closely).

### Procedure

Each participant was tested individually in a quiet laboratory booth. In the first part of the study, participants viewed a sequence of 14 photographs of natural scenes.
The photographs were presented in the center of the computer screen, and each picture was displayed for 10 s. The pictures were presented in the same order for all participants. After each photograph, the seven self-report items were displayed on-screen and participants' responses were collected *via* the keyboard. Before testing began, participants underwent a practice trial with a photograph not used in the main test, to familiarise themselves with the task. Stimulus presentation and response collection were controlled by E-Prime 2.0 Pro.

After the computerised test was completed, participants completed the questionnaires in the following order: Openness/Intellect subscale from the Big Five Aspect Scales, followed by the Connectedness to Nature Scale.

### Data analysis

Aesthetic pleasure and beauty were measured using the pleasing and beauty items, respectively. Aesthetic emotion was measured using the mean of the awe-inspiring, moving, profound and boring (reverse-scored) items (for a similar method, see *Silvia et al. (2015)*).

### Regression models

The first regression model tested whether connectedness to nature and openness predicted aesthetic pleasure evoked by the photographs. The second regression model tested whether connectedness to nature and openness predicted ratings of beauty. The third regression

**Table 1 Descriptive statistics, alphas and zero-order correlations for all measures.**

| Measure | Mean | SD | Alpha | 1 | 2 | 3 | 4 | 5 |
|---|---|---|---|---|---|---|---|---|
| 1. Connectedness to nature | 47.28 | 7.67 | 0.83 | 1 | | | | |
| 2. Openness to experience | 25.79 | 5.59 | 0.71 | 0.43* | 1 | | | |
| 3. Aesthetic pleasure | 5.47 | 0.62 | | 0.45** | 0.30* | 1 | | |
| 4. Beauty | 5.39 | 0.65 | | 0.41** | 0.26 | 0.86** | 1 | |
| 5. Aesthetic emotion | 4.54 | 0.73 | 0.79 | 0.46** | 0.28 | 0.67** | 0.64** | 1 |
| 6. Attention | 5.45 | 0.88 | | 0.31* | 0.13 | 0.40** | 0.35* | 0.26 |

Notes:
To adjust for the multiple correlations that were carried out and control for family-wise error rates, we considered only those correlations to be significant that reached the $p < 0.01$ significance level.
* $p < 0.01$.
** $p < 0.001$.

model tested whether connectedness to nature and openness predicted aesthetic emotion evoked by the photographs.

### Mediation analysis

The potential mediating role of attention in the relationships between connectedness to nature and aesthetic pleasure, and connectedness to nature and beauty was analysed using the PROCESS macro for SPSS (*Hayes, 2018*). This method tests the significance of the indirect effect of the predictor on the outcome through the mediating variable (*Hayes, 2018*). Confidence intervals for the indirect effect that do not contain zero indicate the presence of a significant mediation effect. A full mediation is obtained when the indirect effect is significant and the direct is not, whereas a partial mediation is obtained when both the indirect and direct effects are significant. A precondition to run the mediation analysis was that there were significant correlations between the predictor and the mediator and between the mediator and the outcome variable. Therefore a mediation analysis could not be conducted where the outcome variable was aesthetic emotion, as the correlation between aesthetic emotion (outcome variable) and attention (mediator) did not reach significance.

Potential relationships were examined using a percentile bootstrapped mediation analysis. This technique computed bias-corrected confidence intervals at the 95% level for the indirect effect of the mediator using z = 5,000 bootstrap samples.

## RESULTS

### Descriptive statistics

Table 1 presents descriptive statistics and alphas for connectedness to nature (CN), openness, paying attention, aesthetic pleasure, beauty and aesthetic emotion.
We calculated zero-order correlations for each of the measures. Table 1 shows there were medium correlations between CN and openness, CN and aesthetic pleasure, CN and aesthetic emotion, CN and beauty, CN and attention, openness and aesthetic pleasure, and aesthetic pleasure and attention, openness and aesthetic emotion, and beauty and attention There were small correlations between openness and beauty, and aesthetic emotion and

**Table 2 Results of the regression model for aesthetic pleasure.**

| Predictor | b | β | t | p | Tol. | VIF |
|---|---|---|---|---|---|---|
| (Constant) | 3.60 | | | | | |
| Connectedness to nature | 0.03 | 0.39 | 3.50 | <0.001 | 0.81 | 1.23 |
| Openness to experience | 0.02 | 0.14 | 1.18 | 0.226 | 0.81 | 1.23 |

Note:
b, beta; β, standardised beta; Tol., tolerance; VIF, variance inflation factor.

**Table 3 Results of the regression model for beauty.**

| Predictor | b | β | t | p |
|---|---|---|---|---|
| (Constant) | 3.62 | | | |
| Connectedness to nature | 0.03 | 0.36 | 3.21 | 0.002 |
| Openness to experience | 0.01 | 0.10 | 0.91 | 0.364 |

Note:
b, beta; β, standardised beta.

attention. There were large correlations between aesthetic pleasure, beauty and aesthetic emotion.

## Regression models

### Aesthetic pleasure

We tested whether connectedness to nature and openness predicted ratings of aesthetic pleasure evoked by the photographs. Results indicated that the regression model significantly predicted aesthetic pleasure, ($F(2,79) = 10.75$, MSE = 0.31, $p < 0.001$), accounting for 21.4% of the variance ($R^2 = 0.214$). Connectedness to nature ($p < 0.001$) was a significant predictor of aesthetic pleasure. Openness did not significantly predict aesthetic pleasure ($p = 0.226$) (see Table 2). Table 2 also displays the multicollinearity statistics (Tolerance and Variance Inflation Factor (VIF)) for the model. The values for VIF are all considerably below 10 and the tolerance statistics are all above 0.2. Together these values suggest that there is no multicollinearity in the data (*Field, 2012*).

### Beauty

Next, we tested whether connectedness to nature and openness predicted ratings of beauty. Results indicated that the regression model significantly predicted beauty, ($F(2,79) = 8.41$, MSE = 0.36, $p < 0.001$), accounting for 17.6% of the variance ($R^2 = 0.176$). Connectedness to nature ($p = 0.002$) was a significant predictor of beauty. Openness did not significantly predict beauty ($p = 0.364$) (see Table 3).

### Aesthetic emotion

Here we investigated whether connectedness to nature and openness predicted ratings of aesthetic emotion evoked by the photographs. Results showed that the regression model significantly predicted aesthetic emotion ($F(2,79) = 11.19$, MSE = 0.42, $p < 0.001$), accounting for 22.1% of the variance ($R^2 = 0.22$). Connectedness to nature was a significant

**Table 4 Results of the regression model for aesthetic emotion.**

| Predictor | *b* | *β* | *t* | *p* |
|---|---|---|---|---|
| (Constant) | 2.34 | | | |
| Connectedness to nature | 0.04 | 0.42 | 3.82 | <0.001 |
| Openness to experience | 0.01 | 0.10 | 0.86 | 0.392 |

**Note:**
*b*, beta; *β*, standardised beta.

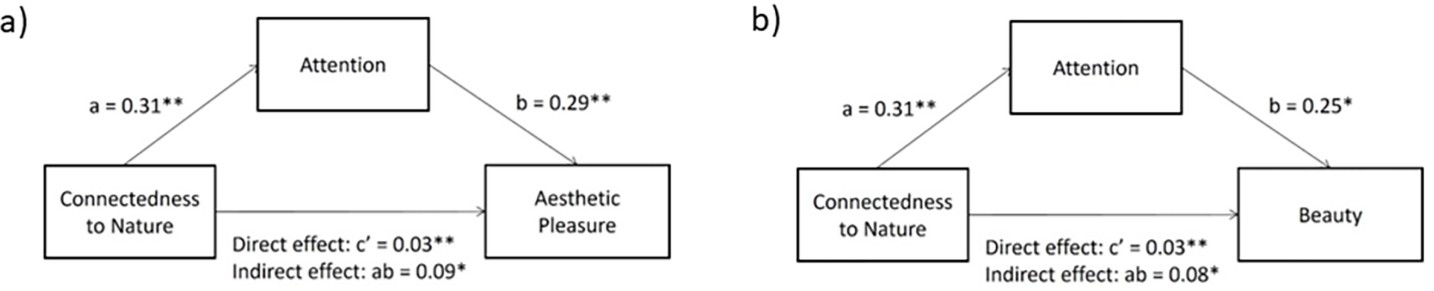

**Figure 1 Mediation models.** Mediation models showing the mediating effect of attention on the relationship between connectedness to nature and aesthetic experience. Standardized coefficients are displayed. *p < 0.05. **p < 0.01.               

predictor of aesthetic emotion ($p < 0.001$), but openness to experience ($p = 0.392$) was not a significant predictor (see Table 4).

## Mediation models

### Aesthetic pleasure

The potential mediating role of attention in the relationships between connectedness to nature and aesthetic pleasure was examined using a percentile bootstrapped mediation analysis (Fig. 1A for a summary of the results). The direct effect of connectedness to nature on aesthetic pleasure was significant ($p < 0.001$ (LLCI = 0.01; ULCI = 0.05)). The indirect effect is significant and mediation can be said to occur if zero falls outside of the 95% confidence interval. The confidence interval for the indirect effect (0.001, 0.019) excluded zero, which is evidence of a significant indirect mediation effect. Therefore, we can conclude that attention partially mediated the relationship between connectedness to nature and aesthetic pleasure.

### Beauty

The potential mediating role of attention in the relationships between connectedness to nature and beauty was examined using a percentile bootstrapped mediation analysis (Fig. 1B for a summary of the results). The direct effect of connectedness to nature on beauty was significant ($p = 0.002$ (LLCI = 0.01; ULCI = 0.05)). The indirect effect is significant and mediation can be said to occur if zero falls outside of the 95% confidence interval. The confidence interval for the indirect effect (0.002, 0.018) excluded zero, which is evidence of a significant indirect mediation effect. Therefore, we can conclude that attention partially mediated the relationship between connectedness to nature and ratings of beauty.

## DISCUSSION

There is a great deal of evidence showing the importance of the appreciation of nature's aesthetic qualities for beneficial outcomes associated from exposure to nature, such as well-being and pro-environmental behaviours (*DeLucio & Mugica, 1994*; *Capaldi et al., 2017*; *Zhang, Howell & Iyer, 2014*; *Zhang et al., 2014*). However, relatively few studies have explored the role of psychological traits in relation to aesthetic responses to the natural world. The aim of the current study was to investigate the relationship between connectedness to nature and participants' aesthetic experiences (including both shallow and deep hedonic tone, and aesthetic judgement) evoked by viewing photographs of natural scenes, and in particular to investigate the role of attention as a mediator in this relationship. Firstly, we found that connectedness to nature positively predicted aesthetic pleasure, beauty and aesthetic emotion, independently of participants' level of openness to experience. Secondly, we found that the amount of attention participants paid to the images mediated the relationship between connectedness to nature and aesthetic pleasure, and between connectedness to nature and ratings of beauty.

As predicted, connectedness to nature was positively associated with aesthetic pleasure, beauty and aesthetic emotion, independently of openness to experience. This finding is in line with studies that have reported a positive association between connectedness to nature and a general disposition to experience natural beauty, as measured using *Diessner et al.'s (2008)* Engagement with Natural Beauty scale (*Harrison & Clark, 2020*; *Zhang, Howell & Iyer, 2014*). Our results extend these findings by measuring the aesthetic responses of participants in the moment, rather than using retrospective self-report measures which rely on episodic and semantic memories. We found that viewers with higher levels of connectedness to nature found the images more pleasant, more beautiful, and they evoked enhanced aesthetic emotions. This finding adds to a growing body of work showing that the trait of connectedness to nature influences perceptual and emotional responses to natural scenes (*Berto et al., 2018*; *Clayton, 2021*; *Davis & Gatersleben, 2013*; *McMahan et al., 2018*; *Tang, Sullivan & Chang, 2015*), but extends our understanding of this association by investigating several specifically aesthetic responses, namely pleasure, beauty and aesthetic emotion. The positive association between connectedness to nature and participants' aesthetic responses to the images of nature is consistent with theories of aesthetic experience which emphasise the importance of the psychological resonance between the viewer and the content of the image (see *e.g.*, *Berleant, 1992*).

In addition, the current study investigated the association between connectedness to nature and the amount of attention paid to the images, as measured by a self-report rating scale. As predicted, we found that participants higher in connectedness to nature paid more attention to the images compared to participants lower in connectedness to nature. This finding is consistent with evidence showing that viewers with higher levels of connectedness to nature allocate more attention to natural scenes during viewing, compared to viewers with lower levels of connectedness to nature, as measured by eye tracking metrics (*Bingjing, Chen & Shuhua, 2022*; *Wu et al., 2013*).

Importantly, we showed that the viewer's level of attention mediated the relationship between connectedness to nature and ratings of aesthetic pleasure and beauty. This is a novel finding, showing that the level of attention paid to the images was a mechanism through which connectedness to nature influenced these aesthetic responses to the pictures. In other words, these findings indicate that nature-connected individuals might experience enhanced aesthetic pleasure and beauty in natural scenes because they paid increased amounts of attention to the scenes. It could be that individuals high in connectedness to nature have an increased drive to acquire environmental information from a scene, therefore are motivated to pay it more attention. This notion is supported by evidence from *Tang, Sullivan & Chang (2015)*, who showed that participants with a higher degree of connection to nature reported higher ratings of mystery (*i.e.*, encouragement to explore the setting and gain further knowledge) in response to viewing natural scenes.

Paying more attention to the scene likely enabled the viewers to extract more detailed information about important features or compositional properties of the scene, leading to enhanced evaluation of the scene's aesthetic qualities. For example, increased attention may have facilitated the recognition and identification of salient features (*e.g.*, trees, water, *etc.*), properties (*e.g.*, colour, form *etc.*), and/or compositional aspects (*e.g.*, harmony, balance *etc.*) that contributed to the scene's pleasantness and beauty. This notion is in line with empirical findings showing that viewers who allocated more visual attention to artworks (as indexed by viewing time) reported enhanced aesthetic appreciation (*Brieber et al., 2014*; *Palumbo et al., 2023*) and tended to rate natural scenes as being more restorative (*van den Berg, Joye & Koole, 2016*). The findings are in conceptual agreement with the results of the mediation analyses reported by *Palumbo et al. (2023)* showing that visual exploration (measured using eye-tracking) mediated the influence of individual differences in personality and cognitive style on aesthetic responses, suggesting that visual exploration is a key mechanism in aesthetic responses to a wide range of stimuli (artworks in a gallery, photographs of nature *etc.*).

Interestingly, increased attention to natural scenes in individuals high in connectedness to nature may also explain the enhanced negative responses to viewing scenes of degraded landscapes by these types of individuals as reported in the study by *Clayton (2021)*.
It would be worthwhile in future studies to investigate the amount of attention paid to nature scenes and its influence on the evaluation of the scene using more implicit indices of attention, such as eye-tracking metrics which could address additional questions about the nature of the enhanced attention to the scenes, for example whether enhanced attention is reflected only in increased viewing times, or if it is also associated with increased spatial exploration of the scene.

Participants who paid more attention to the images reported higher levels of aesthetic pleasure and beauty, but not higher levels of aesthetic emotion. The reason why participants' level of attention was not associated with ratings of aesthetic emotion is unclear, although it may be that evaluations of aesthetic emotion were based primarily on an early 'gist' perception of the entire scene, rather than on more attentionally demanding evaluative processes that unfolded over a longer time-period. Clearly however, further research is needed to understand this more fully.

Openness to experience, which is a personality trait reflecting curiosity and receptivity to new experiences and is a reliable predictor of aesthetic responsiveness (*Chamorro-Premuzic et al., 2009*; *McCrae, 2007*; *Palumbo et al., 2023*; *Silvia & Nusbaum, 2011*), was positively correlated with aesthetic ratings of the scenes. This is in line with studies that showed that participants higher in openness to experience reported increased engagement with natural beauty (*Harrison & Clark, 2020*; *Zabihian & Diessner, 2016*). Importantly however, in the current study, openness to experience was not a significant predictor of aesthetic pleasure, beauty or aesthetic emotions, once connectedness to nature was taken into account.

Several potential limitations of the current study should be mentioned. Our analyses are correlational, so the causal role of connectedness to nature in aesthetic experience cannot be determined. Although the findings showed that connectedness to nature predicted aesthetic pleasure and aesthetic emotion, it is possible that aesthetic responses to nature could enhance the sense of belongingness to nature, thereby increasing connectedness to nature. Further research studies using experimental approaches (perhaps manipulating the level of connectedness to nature) are therefore needed to elucidate the potentially bi-directional causal relationships between these constructs. It should be noted that viewers evaluated images of natural scenes, rather than real natural landscapes. While doubts have been raised regarding the appropriateness of representations instead of real landscapes (*e.g.*, *Hull & Stewart, 1992*), nevertheless there is substantial evidence of the appropriateness of using depictions of natural environments as a substitute for field-based studies (*e.g.*, *Stamps, 1990*). Generalizing the findings of the study should be done with caution due to the relatively small sample size ($N = 82$), and the possibility of sequencing effects from presenting all the measures in the same order (ratings of photos followed by Openness to Experience, followed by Connectedness to Nature) cannot be ruled out. Further studies should include measures relating to the naturalness of one's neighborhood, and degree of exposure to nature in daily life, as additional control variables. Finally, attention was measured here by self-report; further studies could preferentially use more objective measures of attention, for example eye tracking techniques (see *e.g.*, *Berto, Massaccesi & Pasini, 2008*; *Wu et al., 2013*).

## CONCLUSIONS

The aim of this lab-based study was to investigate whether individual differences in the degree of connectedness to nature were related to aesthetic appreciation of photographs of scenes from nature, and to explore the role of attention in this potential relationship. We hypothesized that participants higher in connectedness to nature would rate images of natural scenes as more pleasing, beautiful and emotional compared to participants lower in connectedness to nature. We also hypothesized that the level of attention paid to the scenes would mediate the association between connectedness to nature and aesthetic responses. We found that connectedness to nature positively predicted pleasure, beauty and aesthetic emotion, independently of participants level of openness to experience. We also found that the amount of attention participants paid to the images mediated the relationship between connectedness to nature and aesthetic pleasure, and connectedness to nature and beauty

ratings. These findings extend our understanding by showing that attention is an important mechanism through which connectedness to nature influences aesthetic responses of pleasantness and beauty in response to natural scenes. Building on these findings, future studies should explore which aspects of visual attention play a role in the connectedness to nature-aesthetic response association, and whether the mediating role of attention is influenced by the type of viewed scene (*e.g.*, urban *vs* natural, or mundane *vs* extraordinary nature). The findings have important real-world implications as greater aesthetic sensitivity to nature is associated with several beneficial psychological outcomes, including increased well-being, prosocial and pro-environmental behaviours. It remains an intriguing open possibility as to whether an intervention could increase a person's level of attention to nature (such as for example, *via* instructions to pay attention more closely), which could lead to enhanced aesthetic appreciation, making more likely the beneficial outcomes associated with exposure to nature.

### Open science statement

This study adheres to the following open science standards: data and materials are publicly available at the time of publication, and all manipulations and measured variables are reported.

### Funding

The author received no funding for this work.

### Competing Interests

Neil Harrison is an Academic Editor for PeerJ.

### Author Contributions

- Neil Harrison conceived and designed the experiments, performed the experiments, analyzed the data, prepared figures and/or tables, authored or reviewed drafts of the article, and approved the final draft.

### Human Ethics

The following information was supplied relating to ethical approvals (*i.e.*, approving body and any reference numbers):

   The Faculty of Science at Liverpool Hope University granted ethical approval to carry out the study (Ethical application number: S-16-11-2017 NJ 003)

### Data Availability

   The data is available at the Open Science Framework: Harrison, Neil R. 2023. "Aesthetic Responses to Photos of Nature and NC." OSF. January 4. DOI 10.17605/OSF.IO/WPXCN.

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
