# Peer review of "Level of attention mediates the association between connectedness to nature and aesthetic evaluations of photographs of nature"

_PeerJ, doi:10.7717/peerj.14926_

## Round 0.1 · original submission · Major Revisions

Thank you for your submission, the reviewers have made a number of suggestions for revision that should be addressed.

·

Basic reporting

English language usage is excellent. Background/problem statement is present. The literature review is relevant and sufficient. The structure of the article is fully satisfactory. The figures are meaningful and decent quality. The raw data is available at OSF.

Experimental design

The research is original. The research questions well defined, and the hypotheses are relevant and testable. This study does describe how it fills an identified knowledge gap. The study is rigorous and meets human subjects ethical standards. The methods are described with sufficient detail to replicate.

Validity of the findings

The methodology was appropriate to investigate the hypotheses, and the statistical analysis is sound and appropriate. The conclusions are well stated, relate to the hypotheses, and do not extend beyond the data and results.

Additional comments

Suggestions for minor revisions:
1. 82 participants is a relatively small N. I suggest making a note under Limitations in Discussion that due to sample size, generalizing the findings of the study should be done with caution.
2. Under the Participants subsection you mention “females…males.” Although we do not need to be a slave to APA style, current wisdom indicates that if we are focusing on gender, we should use “woman” and “man” (and perhaps “other”); female and male terminology is used when interested in birth sex, as opposed to gender.
3. I suggest moving your explanation of the 7-self report items to under Stimuli and Measures, as they are measures. Make it clear in that subsection that you are measuring pleasing with 1 item, and beauty with 1 item. Report the alpha for the 3-item aesthetic emotions scale. It appears you did not use the “boring” item; so you should probably mention why not (I assumed you would reverse-score it and include in the aesthetic emotions scale).
Line 253: when I read “pleasing and beauty rating scales” I got confused, as I thought that meant you had a “scale” for each of those (a multiple item scale). Although technically 1 item could be called a scale if it is Likert measured, I think it would be best to refer to those as “pleasing and beauty items”. Same on line 254, I recommend using “items” instead of “scales.”
4. Under Limitations you might mention the possibility of sequencing effects from presenting all the measures in the same order (Emotes > Openness > CN), instead of randomizing them; and/or give your rationale for not randomizing them.
5. Table 1. As you have several correlations in that table, I recommend using a Bonferroni type adjustment for the p values.
6. Lines 338 and 340. I think you meant the indirect effect is “not” significant; and the CI “included” zero.
7. line 362; perhaps when you mention the Engagement with Natural Beauty scale, you should cite the author(s) of the scale too?

Typos:
A. line 167, “tress”
B. line 367 delete “and”
C. line 397 eg = e.g.

I enjoyed reading this paper! It was interesting and meaningful.

·

Basic reporting

This paper is a correlational study that tests if nature connectedness can predict aesthetic appreciation of nature images, mediated by attention to the nature images.

Overall, the basic reporting throughout the paper is very clear and the article is well-structured. All necessary background information is provided to understand the contents of the paper. All figures are relevant and clear. The research question, and how it builds on past research is explained appropriately – in other words, sufficient background information is provided.

Experimental design

Methods are described in a detailed and in an understandable way.

Still, I have some questions and concerns that you might address in a revision of your work.

Why did you exactly choose to control for openness to experience, and not for other traits that could correlate with the aesthetic measures? For me, this appeared to be quite an ad hoc move, as many other measures could be controlled for (e.g., the naturalness of one’s neighborhood).

I also wonder whether the research question is not a bit too obvious. For me, it appears to be expected that people who feel connected to nature will also positively evaluate stimuli that are central to their personality. Why should we think otherwise?

Validity of the findings

Regarding validity, I was wondering whether connectedness and positive aesthetic judgment do not tap into the same underlying construct and whether this explains your association.

I am not super-convinced about the mediation analyses. What’s your rationale for picking attention as a mediator – the choice for attention came across as a bit “random”. This feeling got strengthened by the fact that you measure attention with only one self-report item, which might raise concerns about validity.

Further, given that your setup is correlational, a critical interpretation of your mediation findings is that they just show correlations between a host of variables. I am not sure how to solve this, and whether it can be solved at all, and I also realize that without the mediation analyses the paper will lose quite a bit of substance.

Additional comments

Could you please include an Open Science statement?

Looking at the mediation figures, I see that non-standardized coefficients are reported. Perhaps, for ease of interpretability, it is a good idea to provide standardized beta’s (obtained by standardizing all variables in the analyses).

Overall, I genuinely enjoyed reading the paper; the paper is very fluently written, the author shows great scholarship and I could not detect any big issues in the paper. That being said, I was expecting some more substance to the paper. Perhaps an experimental design could have been more appropriate, or at least provide an interesting complementary perspective. Would an additional study where feelings of connectedness are manipulated not give you a stronger package? If feasible, I would motivate you to do so.

Best of luck!

Reviewer 3 ·

Basic reporting

Level of attention mediates the association between nature connectedness and aesthetic evaluations of photographs of nature

Abstract
-It would be good to make clear the study design here – pre post, random assignment, or more survey based with photographic exposure for all participants?

Intro
Line 81-82, I would cite something more recent in addition to Ulrich here
“Empirical studies in environmental aesthetics have tended to investigate aesthetic
preferences for different types of landscape, for example forests and savannahs (Heerwagen &
Orians, 1993) and how such preferences are associated with the landscape’s formal and structural properties (for review, see Ulrich, 1983).”

Language: nature connectedness, and connectedness to nature, and inclusion of nature in self are related but distinct in how they’re measured and described in some ways – it would be good to keep the language consistent as to which you are referring to in the introduction

Correction: Davis and Gatersleben (2013) found that higher levels of nature connectedness predicted more positive emotional experiences – it would be more accurate to say “Davis and Gatersleben (2013) found that higher levels of nature connectedness predicted more positive emotional experiences - both high and low arousal emotions (exciting and calming).” It was also found that those high in CN had less negative or fearful experiences in more extreme/scary natural environments providing a sort of buffering effect.

BASIC REPORTING
Clear, unambiguous, professional English language used throughout. YES

Intro & background to show context. YES

Literature well referenced & relevant. YES

Structure conforms to standards, discipline norm, or improved for clarity. YES

Figures are relevant, high quality, well labelled & described. YES

Experimental design

EXPERIMENTAL DESIGN
• Original primary research within Scope of the journal. YES, it is original and helpful.
• Research question well defined, relevant & meaningful. It is stated how the
research fills an identified knowledge gap. YES, though making clear it is correlational earlier would be helpful.
• Rigorous investigation performed to a high technical & ethical standard. YES.
• Methods described with sufficient detail & information to replicate. YES.

Validity of the findings

VALIDITY OF THE FINDINGS
• Impact and novelty not assessed. NOT entirely novel but important and a contribution.
• Meaningful replication encouraged where rationale & benefit to literature is clearly
• stated. YES.
• All underlying data have been provided; they are robust, statistically sound, &
• controlled. YES.
• Conclusions are well stated, linked to original research question & limited to
supporting results. YES, though more information on the impact /real world implications of this – for health, nature exposure, etc would be helpful, especially in the abstract a bit and the conclusion.

Additional comments

One of the easier articles I've had to read and digest in a while - well written and clear in its intentions and abiding to standards of research and reporting. I am sorry for not providing a more detailed review but I've had a low grade covid fever for a week, despite my request for an extension. I hope my review is sufficient - I believe the paper is worthy of publication nearly as it is.

---

## Round 0.2 · accepted · Accept

Thank you for your revision - I am please to tell you this article has been accepted for publication.

·

Basic reporting

The revision is good. Good to go.

Experimental design

x

Validity of the findings

x

Additional comments

x

·

Basic reporting

No comments

Experimental design

No comments

Validity of the findings

No comments

Additional comments

Perhaps you could move up the "Open Science Statement" a little bit more up in the text, rather than placing it at the end.